# Research Progress on the Mechanism of Anti-Tumor Immune Response Induced by TTFields

**DOI:** 10.3390/cancers15235642

**Published:** 2023-11-29

**Authors:** Yue Lan, Shaomin Zhang, Yun Pan, Minmin Wang, Guangdi Chen

**Affiliations:** 1Department of Public Health, Zhejiang University School of Medicine, Hangzhou 310027, China; lanyue@zju.edu.cn; 2Qiushi Academy for Advanced Studies, Department of Biomedical Engineering, Zhejiang University, Hangzhou 310027, China; shaomin@zju.edu.cn; 3Key Laboratory of Biomedical Engineering of Ministry of Education, Zhejiang University, Hangzhou 310027, China; 4Zhejiang Provincial Key Laboratory of Cardio-Cerebral Vascular Detection Technology and Medicinal Effectiveness Appraisal, Zhejiang University, Hangzhou 310027, China; 5College of Information Science and Electronic Engineering, Zhejiang University, Hangzhou 310027, China; panyun@zju.edu.cn; 6Binjiang Institute of Zhejiang University, Zhejiang University, Hangzhou 310027, China

**Keywords:** tumor treating fields, mechanism of action, anti-tumor immunity, immunotherapy

## Abstract

**Simple Summary:**

Although conventional and emerging treatment modalities have improved the survival outcomes of cancer patients to some extent, the prognosis for solid tumors remains notably grim. Tumor treating fields, as a novel approach for cancer therapy, have shown promising effects as both a monotherapy and in combination with chemotherapy. However, the underlying mechanisms through which TTFields exert their anticancer effects remain incompletely understood. In recent years, many researchers have suggested that inducing anti-tumor immune responses may be one of the key mechanisms of the anticancer activity of TTFields. Several clinical trials are currently exploring the combination of TTFields with tumor immunotherapy and have achieved positive results. This article reviews the research progress on the mechanisms of TTFields-induced anti-tumor immune responses and discusses their clinical significance. The objective is to offer a streamlined and precise summary to facilitate the future exploration and advancement of TTFields.

**Abstract:**

Tumor treating fields (TTFields), a biophysical therapy technology that uses alternating electric fields to inhibit tumor proliferation, has been approved by the U.S. Food and Drug Administration (FDA) for the treatment of newly diagnosed or recurrent glioblastomas (GBM) and malignant pleural mesotheliomas (MPM). Clinical trials have confirmed that TTFields are effective in slowing the tumor growth and prolonging patient survival. In recent years, many researchers have found that TTFields can induce anti-tumor immune responses, and their main mechanisms include upregulating the infiltration ratio and function of immune cells, inducing the immunogenic cell death of tumor cells, modulating immune-related signaling pathways, and upregulating the expression of immune checkpoints. Treatment regimens combining TTFields with tumor immunotherapy are emerging as a promising therapeutic approach in clinical practice. Given the increasing number of recently published studies on this topic, we provide an updated review of the mechanisms and clinical implications of TTFields in inducing anti-tumor immune responses. This review not only has important reference value for an in-depth study of the anticancer mechanism of TTFields but also provides insights into the future clinical application of TTFields.

## 1. Introduction

Tumor treating fields are a novel, non-invasive anti-tumor therapy that delivers a low-intensity (1–3 V/cm), medium-frequency (100–300 kHz) alternating electric field to the tumor through an electrodes array covered on the skin. It was reported that the alternating electric field causes positively and negatively charged microtubulins with dipole moments in tumor cells to rotate and align on opposite electrodes, thereby interfering with the directional movement and assembly of microtubule proteins [1]. Microtubulin is an important component of the spindle, and the action of TTFields on microtubulin disrupts the assembly of the spindle and the formation of the cleavage furrow in the mitotic process of tumor cells, thus preventing their division and proliferation [2,3]. Many preclinical studies and clinical trials demonstrated that TTFields can inhibit the growth and progression of a wide range of solid tumors, such as glioblastoma, non-small cell lung cancer, mesothelioma, hepatocellular carcinoma, ovarian cancer, and so on [4,5,6,7] (Figure 1).

In April 2011, the U.S. Food and Drug Administration (FDA) granted approval for the utilization of TTFields in the therapeutic management of glioblastoma [8]. Glioblastoma multiforme (GBM) is the primary malignant brain tumor with the highest incidence and worst prognosis in the adult population. The prevailing clinical protocol for GBM entails surgical intervention coupled with radiotherapy and temozolomide chemotherapy, with a median survival of 16–17 months and a five-year survival rate of only 9.8% for patients after treatment [9,10]. The collective findings of several clinical trials unequivocally underscored the efficacy of TTFields in conferring a survival advantage to patients afflicted with GBM [11,12]. According to the results of a phase III clinical trial targeting newly diagnosed glioblastoma patients, NovoTTF-100A (developed by Novocure) combined with temozolomide adjuvant chemotherapy extended progression-free survival from 4.0 months to 7.1 months, compared to chemotherapy alone [13].

TTFields have demonstrated a superior safety profile in the treatment of GBM patients. The limited number of non-cancerous dividing cells in the brain greatly reduces the impact and side effects of TTFields on normal cells. This aspect renders TTFields potentially safe for use, an assertion supported by the results of a phase III clinical trial involving the NovoTTF-100A device. This trial demonstrated the effectiveness of TTFields in the treatment of GBM compared to the chemotherapy option known as the Best Practitioner’s Choice (BPC). Furthermore, chemotherapy can lead to secondary systemic infections, immune system disruption, or demyelination changes in some patients, whereas TTFields have fewer undesired side effects [14].

Several clinical trials have been or are exploring the efficacy of TTFields in various solid tumors. The results of a randomized phase III clinical trial conducted by Stupp et al. in October 2015 showed that, compared to chemotherapy alone, the combination of TTFields and chemotherapy extended the median overall survival of patients by 4.9 months [15]. Data from phase II clinical trials showed that the combination of TTFields and pemetrexed chemotherapy improved the overall survival rate of malignant pleural mesothelioma patients and effectively slowed down tumor progression within the patients’ bodies [12]. Interestingly, compared to patients receiving chemotherapy alone, patients receiving TTFields showed significant improvements in their quality of life, emotional well-being, physical functioning, and cognitive abilities [16]. Due to the effectiveness of TTFields, clinical trials combining TTFields with standard chemotherapy are currently being conducted for many other types of cancer. The above results suggest that TTFields may potentially have advantages in the clinical treatment of many solid tumors in the future.

Although TTFields have shown bright prospects in clinical applications, the molecular mechanism of TTFields is still unclear. Currently, research into the anticancer mechanisms of TTFields mainly focuses on their effects on the mitotic process of tumor cells. Some researchers believe that alternating electric fields primarily exert their anticancer effects by influencing the mitosis of tumor cells. TTFields can interfere with the attachment of sister chromatids to the spindle fibers and disrupt the stability of microtubule protein heterodimers [17]. In addition to regulating spindle assembly and orientation, electric fields can also interact with proteins with high dipole moments. For example, TTFields can inhibit the localization of the Septin protein complex, which is crucial for the formation and positioning of the cytokinetic furrow during cell division [18]. The regulation of the Septin complex by TTFields can lead to abnormal cytokinetic furrows, causing tumor cells to undergo defective mitosis and subsequent cell death [19].

Although the mechanism described above provides an explanation for many of the anti-tumor effects of TTFields, it is not a comprehensive one that can account for all the observed phenomena. There are still questions that remain unanswered, such as why TTFields only interfere with the mitosis of tumor cells, while having little impact on normal cells, and why different types of tumor cells exhibit varying intensity- and frequency-sensitivity to TTFields. The intervention frequency of alternating electric fields produces varying effects on tumor cells and normal cells. However, this viewpoint still needs more experimental data support. In short, it is currently unclear what characteristics differentiate tumor cells from non-tumor cells and make tumor cells more vulnerable to TTFields.

Many experts in oncology remain skeptical of TTFields due to the lack of a precise molecular mechanism and sufficient data of clinical application, making it difficult to maximize the therapeutic potential of TTFields in the clinic [20,21]. In recent years, more and more studies have speculated that TTFields may be capable of inducing specific anti-tumor immune responses for its anti-tumor effects [22], which may answer why TTFields specifically attack tumor tissues. Moreover, exploring the correlation between TTFields and tumor immunotherapy may enhance the anti-tumor effects of TTFields. For example, the therapeutic regimen of TTFields combined with immune checkpoint inhibitors (ICIs) is about to enter the clinical trial stage. Therefore, this article reviews the latest research progress on the mechanism of the TTFields-induced anti-tumor immune response and looks forward to the clinical significance in this field, with a view to providing a reference for the clinical application of TTFields in the field of tumor immunotherapy.

## 2. Mechanism of Anti-Tumor Immune Response Induced by TTFields

Local and systemic immune suppression is indeed an important characteristic of GBM. There is an abundance of immune suppressive cells in the tumor microenvironment of GBM, including myeloid-derived suppressor cells (MDSCs) and regulatory T cells (Tregs). The presence of these cells can inhibit the effective response of the immune system against the tumor, thereby promoting tumor growth and spread [23]. This immunosuppression is one of the main reasons for the unfavorable prognosis encountered by GBM patients subject to conventional therapeutic paradigms. Therefore, current therapeutic regimens for GBM are conspicuously centered on reversing immunosuppression, a conceptual analogy akin to the transformation of a ‘cold’ tumor into an ‘inflamed’ or ‘hot’ tumor milieu. Several studies have proffered intriguing implications of Tumor Treating Fields (TTFields) therapy in this regard. It was reported that TTFields treatment in combination with temozolomide is effective in prolonging the median survival of GBM patients compared to chemotherapy alone [14]. Notably, an exploration into the differentially expressed gene profile of glioblastoma cell lines prior to and post TTFields intervention unveiled a substantive modulation of 654 genes germane to immune and inflammatory cascades [24]. The above results suggest that TTFields may exert anticancer effects by altering the inflammatory environment and the immune system, reversing the immunosuppression in the GBM tumor microenvironment.

Researchers conducted a survey of GBM patients undergoing TTFields treatment. The results showed that individuals receiving daily injections of >4.1 milligrams of dexamethasone (n = 64) had a median survival of 4.8 months, whereas individuals receiving daily injections of ≤4.1 milligrams of dexamethasone (n = 56) had a median survival of 11 months (*p* < 0.0001) [25]. Dexamethasone, acknowledged for its capacity to temper the magnitude of the anti-tumor immune response, achieves this by quelling the inflammatory cascade, attenuating the release of pro-inflammatory mediators like tumor necrosis factor TNF-α, and abrogating the proliferation and effector functions of immune cells. Therefore, the effect of high-dose dexamethasone on the prognosis of TTFields-treated patients suggests that the tumor-suppressive effect of TTFields treatment in GBM patients may be dependent on the status of the immune system.

In summary, there is a suggestion that the anti-tumor immune response potentially serves as one of the mechanisms underlying the anti-tumor effects of TTFields. Thus, we present a concise overview of the mechanisms pertinent to the initiation of the anti-tumor immune response by TTFields (Figure 2).

### 2.1. TTFields Enhance Immune Cell Infiltration and Function In Vivo and In Vitro

TTFields upregulated the percentage of lymphocyte infiltration. In animal models of glioma, TTFields demonstrated efficacy in prolonging survival time, suppressing tumor cell proliferation, and fostering tumor cell apoptosis. These outcomes are potentially attributed to the heightened upregulation of CD8+ T cell infiltration within the tumor microenvironment due to TTFields [26]. The results from clinical trials further underscore that TTFields have the capacity to upregulate the proportion of tumor-infiltrating lymphocytes (TILs) infiltrating the tumor tissues within some GBM patient tumors [27].

TTFields enhance the anti-tumor effects of lymphocytes. While TTFields disrupt the mitosis of tumor cells, they appear to have no effect on the normal function of T cells. Following TTFields treatment, the peripheral blood T cells and TIL from GBM patients retained their original anti-tumor functions, including the secretion of interferon (IFN-γ), cytotoxic degranulation, and the expression of programmed death receptor (PD-1) [27]. Moreover, there is emerging evidence that TTFields enhance the secretion of pro-inflammatory cytokines by lymphocytes and also heighten the level of lymphocyte-induced memory immune responses. TTFields cause the release of micronuclei from the nucleus into the cytoplasm by inducing the focal disruption of the nuclear membranes of glioblastoma cells, where the micronuclei activate and recruit the stimulator of interferon genes (STING), which are absent in melanoma2 (AIM2)/caspase1 inflammatory vesicles. This promotes the production of type1 interferons (T1IFNs) and the proportion of CD4+ T cells and CD8+ T cells infiltrated in the tumor cell microenvironment [28].

TTFields enhance lymphocyte immune memory. In a murine glioma model, researchers found that the anti-tumor immune response elicited by TTFields persisted for a span of 3 days following the cessation of treatment. This finding implies that TTFields not only induced an inflammatory response but also the induction of a lasting immune memory targeting tumor antigens [28]. To further investigate the induction of the memory immune response by TTFields in vivo, researchers conducted an experiment where the same tumor cells were re-injected into mice models of glioblastoma that had either received or not received TTFields treatment. The results showed a significant prolongation of median survival in mice that had received TTFields treatment, accompanied by a significant increase in TCRs clonal expansion [28]. Collectively, these results indicate that TTFields may have an activating effect on lymphocyte-mediated specific anti-tumor immune responses and memory immune responses, which is expected to reverse the immunosuppressive nature of the tumor microenvironment [29].

TTFields promote dendritic cell maturation and phagocytosis. Lewis lung cancer cells that have undergone TTFields treatment can enhance the expression of maturation markers (MHC II, CD40, and CD80) on bone-marrow-derived dendritic cells (BMDCs) and improve their ability to engulf tumor cells [30]. These findings indicate that TTFields-intervened tumor cells may promote the mature phagocytosis of DCs by releasing damage-associated molecular patterns that bind to the receptors of DCs.

TTFields enhance tumor killing by macrophages. TTFields induce macrophage RAW264.7 to secrete pro-inflammatory cytokines, such as reactive oxygen species (ROS), the inflammatory mediator nitric oxide (NO), and IL-1β, TNF-α, and IL-6, in vitro, which enhances the killing of mouse mammary adenocarcinoma cells, 4T1 [31]. Consequently, the findings suggest that TTFields activate downstream anti-tumor effects by activating mitogen-activated protein kinase (MAPK) and nuclear factor-kB (NF-kB) signaling pathways in macrophages [32].

### 2.2. TTFields Induce Immunogenic Cell Death in Tumor Cells

In the past, scientists believed that apoptosis was a silent mode of cell death that did not activate the acquired immune response. However, recent research showed that cells undergoing programmed death following bacterial or viral infection release signals, including surface-exposed calreticulin (CRT), damage-associated molecular patterns (DAMPs), and high-mobility-group protein 1 (HMGB1). These signals activate receptors on dendritic cells (DCs), inducing DC maturation, the uptake of tumor antigens, and the subsequent activation of T cells into cytotoxic T lymphocytes (CTLs) with a tumor-killing function, a process known as immunogenic cell death (ICD) [33]. Recent studies demonstrated that many oncology treatments maintain a durable anti-tumor immune response by inducing ICD [33,34].

TTFields induce immunogenic cell death (ICD) in tumor cells. Recent research showed that TTFields induce the release of HMGB1 protein from tumor cells and promote the exposure and high expression of CRT proteins. These characteristic changes in ICD were observed in various tumor cell lines that received TTFields intervention, such as mouse Lewis lung carcinoma (LLC-1), mouse colon carcinoma (CT-26), mouse ovarian surface epithelium (MOSE-L), human hepatocellular carcinoma (HepG2), and human lung squamous cell carcinoma (H520) cells [30,35]. However, the above studies did not confirm the necessity of ICD in TTFields’ anti-tumor effect. Additionally, some studies confirmed that tumor cells undergoing aberrant mitosis give rise to hyperploid cancer cells, which induce ICD in tumor cells and activate the immune system to undergo immunosurveillance and anti-tumor immune responses [36]. Since interfering with tumor cell mitosis was the earliest identified anti-tumor mechanism of TTFields, we speculate that the induction of ICD in tumor cells by TTFields is one of the mechanisms through which TTFields exert their anti-tumor effects.

TTFields may induce ICD in tumor cells by promoting endoplasmic reticulum stress. Tumor cells exposed to TTFields undergo ICD accompanied by the release of the endoplasmic reticulum stress- and damage-associated molecular pattern Alarmin/DAMP [35]. Some researchers suggested that TTFields trigger the exposure of calreticulin by upregulating the phosphorylation of the endoplasmic reticulum stress-related translation initiation factor eIF2α [30]. The above results suggest that endoplasmic reticulum stress may be the mechanism by which TTFields induce ICD in tumor cells.

### 2.3. TTFields Regulate Immune-Related Signaling Pathways

TTFields may exert anti-tumor effects by modulating signaling pathways associated with inflammation and immune responses in tumor cells. TTFields inhibit tumor growth in a GBM mouse model by upregulating miR29b expression in glioma cells, thereby targeting the Akt2 signaling pathway [37]. MiR29b can silence downstream oncogenes and is associated with tumor cell proliferation, apoptosis, and immune regulation [38]. Akt2 is an important oncogene involved in the regulation of several immune-related signaling pathways [39]. The regulation of this pathway by TTFields suggests a regulatory role of TTFields in the tumor immune system. Other researchers demonstrated that TTFields affect the amount and perinuclear localization of U87 microtubule proteins in glioma cells, hindering the targeted assembly of microtubule proteins. This alteration upregulates the level of activated RhoA and its downstream Rho-associated coiled-coil kinase (ROCK) pathway, leading to a disruption of the actin cytoskeleton structure in tumor cells [40]. RhoA is a key regulator of immune cell differentiation and function, participating in DC antigen presentation and the formation of immune synapses between DCs and specific T cells [41,42]. Taken together, TTFields may exert anti-tumor effects by regulating inflammatory and immune-related signaling pathways, suggesting that TTFields play an important regulatory role in anti-tumor immune responses.

### 2.4. TTFields Upregulate Immune Checkpoints

TTFields upregulate the expression of immune checkpoints in tumor cells. Single-cell sequencing results showed that immune checkpoints such as programmed death-ligand 1 (PD-L1), cytotoxic T-lymphocyte-associated protein 4 (CTLA-4), and the T cell immunoglobulin and ITIM domain (TIGIT) are upregulated on the surface of tumor cells after TTFields intervention [28]. It was suggested that the upregulation of PD-L1 in tumor cells after treatment is a result of the activation of the adaptive immune system [43,44].

TTFields combined with immune checkpoint inhibitors may effectively inhibit tumor growth. The results of TTFields combined with immune checkpoint inhibitor anti-PD1 intervention in an in situ Lewis lung cancer mouse model showed that the tumor volume inhibition effect of the combined treatment was significantly higher than that of the single treatment group, and the proportions of macrophages, DCs, CD8+ T cells, and CD4+ T cells in the tumor microenvironment of mice receiving the combined treatment were all upregulated [30]. PD1 is expressed on the surface of T cells, and programmed death-ligand 1 (PD-L1) is expressed on the surface of tumor cells. They form an important pair of immune checkpoints, and their combination can inhibit T cell proliferation and pro-inflammatory cytokine secretion, which is an important target for reversing tumor immunosuppression and drug resistance [45,46]. Other researchers also demonstrated that TTFields combined with anti-PD1/anti-CTLA-4/anti-PD-L1 therapy can significantly reduce tumor volume and increase the infiltration of cytotoxic T lymphocytes and memory CD8+ T cells in the tumor microenvironment [47]. These results provide strong evidence for the upregulation of immune checkpoints by TTFields, suggesting that TTFields may bring new hope for the clinical treatment of tumors through combined tumor immunotherapy in the future.

## 3. Clinical Significance of Anti-Tumor Immune Responses Induced by TTFields

There is an interaction between tumors and the immune system, and the inhibitory effect of the immune system on tumor tissue progression is widely recognized [48]. In addition to preventing normal cells from becoming tumor cells by eliminating the invasion of foreign pathogens, the immune system can activate specific anti-tumor immune responses by recognizing the specific antigenicity expressed by tumor cells, thereby eliminating tumor cells. This process is known as tumor immune monitoring [49]. However, certain neoplastic cells possess the ability to elude the vigilant scrutiny of the immune system, persisting in their development—a phenomenon denoted as tumor immune editing. Tumor immune editing involves elimination, balance, and escape stages. When only a portion of tumor cells is cleared by the immune system, a brief equilibrium state is formed between the immune system and the progressing tumor cells. If the immune system cannot completely eliminate the tumor, or the tumor accumulates enough DNA mutations at this stage, it enters the escape stage where the immune system cannot suppress tumor progression [50]. This highlights the relationship between the immune system and the occurrence and progression of tumors.

Traditional and emerging anti-tumor treatment methods regulate the interaction between tumors and the immune system. Conventional treatment methods, such as chemotherapy, surgery, and radiotherapy, reduce the immunosuppressive properties of tumors by reducing the tumor volume. They can also induce tumor cell necrosis, releasing or exposing hidden tumor antigens. Additionally, chemotherapy can activate the immune system by activating immune-related signaling pathways or regulatory factors [51]. Emerging tumor treatment methods, such as photodynamic therapy and photothermal therapy, induce tumor cell apoptosis and release tumor antigens, leading to the infiltration of white blood cells and the release of pro-inflammatory cytokines in the immune system, promoting the specific killing of tumor cells by cytotoxic T lymphocytes (CTLs) [52,53]. Based on this characteristic, in recent years, radiotherapy, chemotherapy, photothermal therapy, and photodynamic therapy combined with tumor immunotherapy have achieved significant results. Treatment methods such as radiotherapy and chemotherapy directly lead to tumor cell death, comprehensively triggering tumor cells to release antigens, thereby improving the level of anti-tumor immune response and solving the problem of low efficiency in simple immunotherapy. These approaches, when combined with immunotherapy, improve the specificity, universality, and persistence of anti-tumor therapies [54,55]. Since TTFields can also induce apoptosis in tumor cells and induce anti-tumor immune responses, the above results suggest that TTFields may also modulate the interaction between the tumor and the immune system.

TTFields may inhibit tumor tissue metastasis by activating anti-tumor immune responses. Researchers have found that TTFields treatment significantly reduces the number and diameter of lung metastases in a rabbit epidermal squamous cell carcinoma model. TTFields treatment also significantly increased the proportion of CD4+ T cells and CD8+ T cell infiltration inside and around tumor metastases [56]. This suggests that TTFields inhibit tumor metastasis and invasion by activating a systemic immune response targeting tumor antigens at the primary site. Although there is a lack of clear evidence on whether the above processes are directly related to immune system activation, these results imply that TTFields therapy may have the potential to eliminate residual cancer cells or inhibit the invasion of small metastatic lesions. Therefore, the local or systemic immune system activation that occurs during the TTFields treatment of in situ tumors may assist in tumor eradication and inhibit tumor metastasis before, during, and after surgery. In summary, studying the role and mechanism of TTFields in anti-tumor immune response can provide a wider range of therapeutic applications for TTFields in clinical practice.

TTFields combined with tumor immunotherapy improves anti-tumor efficacy. At the 2023 American Society of Clinical Oncology (ASCO) annual meeting, a phase III clinical study evaluating the use of TTFields in combination with standard therapy for non-small cell lung cancer (NSCLC) was presented. The results showed that the combination of TTFields and immune checkpoint inhibitors increased the median survival of patients by 8 months. The previous Section 2.4 suggests that the combination of TTFields and immune checkpoint inhibitors can improve the level of the anti-tumor immune response in the tumor microenvironment. These results indicate that TTFields may improve the exposure and expression of tumor antigens, enhance the anti-tumor efficiency of the immune system, and have promising clinical application prospects, similar to traditional tumor treatment methods. However, the specific mechanism by which TTFields enhance the level of anti-tumor immune response is still unclear.

## 4. Conclusions and Prospect

In conclusion, the results of in vivo, in vitro, and clinical trials confirm the activation and enhancement effect of TTFields on anti-tumor immune response. This article provides the first review of the relevant mechanisms of TTFields activating anti-tumor immune responses. The main mechanism of TTFields inducing an anti-tumor immune response includes upregulating the number and function of immune cells, promoting the release and exposure of tumor cell antigens, and regulating immune-related signaling pathways and immune checkpoint inhibitors. Based on these mechanisms, more and more combination therapy schemes are being explored in clinical research. However, due to the lack of clear molecular mechanisms, determining an optimal combination therapy regimen remains challenging.

Clarifying the exact molecular mechanism of TTFields in anti-tumor therapy is expected to overcome the problem of the low sensitivity of some patients to radiotherapy and chemotherapy. For instance, research showed that TTFields can effectively reduce the survival rate of human breast cancer multidrug resistance (MDR) cell lines that overexpress ATP-binding cassette (ABC) transporters [57]. The overexpression of ABC transporters in tumor cells can lead to decreased drug concentration and contribute to resistance to chemotherapy drugs. While chemotherapy sensitizers that block ABC transporters have not achieved significant results in clinical practice [58], the application of TTFields can normalize the concentration of chemotherapy drugs in drug-resistant cells overexpressing ABC transporters, similar to wild-type tumor cells [57]. These findings suggest that the molecular mechanism by which TTFields exert anti-tumor effects may directly or indirectly inhibit the expression and function of ABC transporters. Thus, understanding the specific molecular mechanisms of TTFields in anti-tumor therapy can contribute to its wider clinical application.

The combination of TTFields and immune checkpoint inhibitors has achieved unprecedented therapeutic effects in clinical practice. Therefore, further research into the molecular mechanisms of TTFields in inducing anti-tumor immune responses is crucial. For example, in Section 2.2, we stated that TTFields have the ability to induce immunogenic cell death in tumor cells. Although tumor cells treated with TTFields exhibit characteristics of immunogenic cell death, the specific mechanism by which TTFields induce this complex immune response has not been fully understood. Some unresolved questions include which receptors on dendritic cells recognize the signals released by tumor cells, how dendritic cells process engulfed tumor antigens, how T cells are activated, and whether the activated cytotoxic T lymphocytes can effectively kill tumors. Research into these questions can help identify patients who are more suitable for TTFields treatment and enhance the activity of relevant signaling pathways to improve therapeutic outcomes. For instance, Lionel et al. discovered that during radiotherapy and chemotherapy, dendritic cells process and present tumor antigens to T cells through the pattern-recognition receptor TLR4 and its downstream MyD88 signaling pathway [59]. Based on this principle, researchers found that patients with low TLR4 gene expression levels had a worse prognosis after radiotherapy and chemotherapy.

The activation of the immune system not only plays a pivotal role in specifically inhibiting the proliferation and progression of tumor cells but also is intricately associated with phenomena such as drug resistance, micrometastasis, and various other intricate processes. Some researchers suggest that the limited electric field coverage of TTFields is a drawback, and the activation of the local or systemic immune system may overcome this limitation. Enhancing the activity of TTFields as a target for immune system activation can potentially induce a more long-lasting and localized tumor-killing effect. However, the current research into TTFields inducing an anti-tumor immune response mainly focuses on observing the changes in immune cells before and after TTFields treatment. The exploration of the correlation between TTFields and the above changes is limited. Researchers should pay more attention to the correlation between TTFields and immune response. In summary, the prospects of TTFields in the clinical treatment of tumors hold great promise, and further research into their role in anti-tumor immune responses is needed.

## Figures and Tables

**Figure 1 cancers-15-05642-f001:**
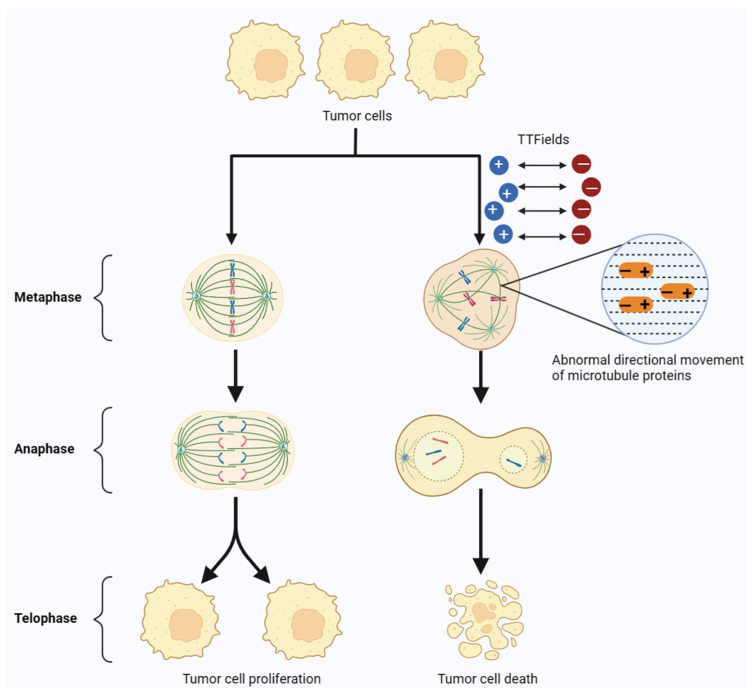
TTFields model for interfering with tumor cell mitosis. In the anaphase of tumor cell mitosis, TTFields can interfere with the formation and directional movement of microtubulin, ultimately leading to the apoptosis of tumor cells.

**Figure 2 cancers-15-05642-f002:**
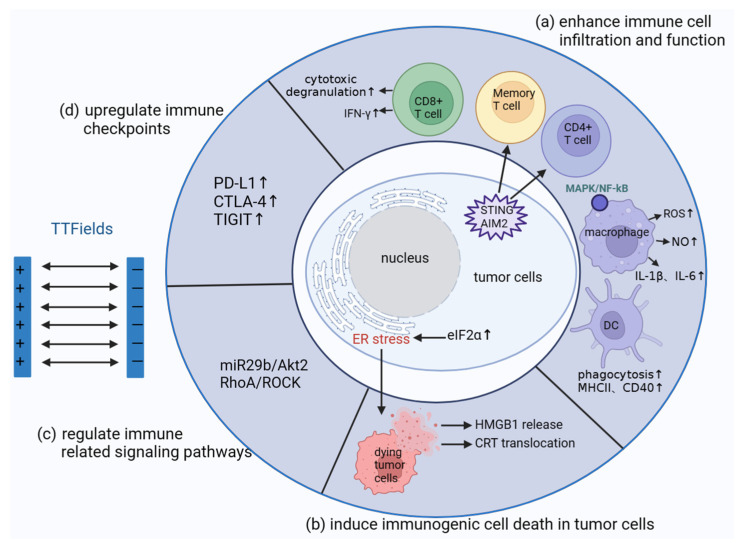
The potential mechanisms of TTFields inducing anti-tumor immune response. (**a**) The mechanism of TTFields enhancing immune cell infiltration and function; (**b**) the process of immunogenic cell death induced by TTFields in tumor cells; (**c**) TTFields have a regulatory effect on inflammation and immune-related signals in tumor cells; (**d**) TTFields enhance anti-tumor immune response by upregulating immune checkpoints in tumor cells. ↑, means that the level of cytokines secreted by cells increases or the expression level of checkpoints on the cell surface increases.

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
