# Peer review of "Research Progress on the Mechanism of Anti-Tumor Immune Response Induced by TTFields"

_cancers, 2023, doi:10.3390/cancers15235642_

Round 1
Reviewer 1 Report
Comments and Suggestions for Authors
This is a review of mostly preclinical data on the potential anti-tutor immune response induced by TTFields.
This is a well written paper that reviews the subject extensively.
Most of the data is pre-clinical but it is well presented and easy to follow.
There is minor editing:
page 3 line 79: there is a missing coma after cells.
Page 3 line 86: the sentence starting with "In addition seems to be misplaced
Author Response
Dear Reviewer:
Thanks very much for taking the time to review this manuscript. Those comments are all valuable and very helpful for revising and improving our paper. We have studied your comments carefully and have made revisions which are highlighted in the paper. The main corrections in the paper and the responses to your comments are as follows:
- Response to comment: (This is a review of mostly preclinical data on the potential anti-tutor immune response induced by TTFields.This is a well written paper that reviews the subject extensively. Most of the data is pre-clinical but it is well presented and easy to follow.)
Response: Thank you for your positive comments.
- Response to comment:( page 3 line 79: there is a missing coma after cells.)
Response: Thank you for the constructive comments. We have incorporated a period at the end of the sentence to signify its conclusion.
- Response to comment:(page 3 line 86: the sentence starting with "In addition seems to be misplaced.)
Response: Thanks for your suggestions. Following your advice, we have revised this sentence to read as follows: Several clinical trials are exploring the efficacy of TTFields in various solid tumors. The purpose of this sentence is to state that there are many completed or ongoing clinical trials exploring the inhibitory effect of TTFields on solid tumors.

Reviewer 2 Report
Comments and Suggestions for Authors
First, my congratulations and appreciation on putting together an exhaustive and comprehensive review.
Some suggestions:
1. In the plain summary section, it would be better to say "anticancer activity of TTFields" rather than "TTFields ’ anticancer activity
2. Abstract- Authors call it " physical therapy technology"- is better called biophysical, physical therapy means something else to a clinician.
3. "TTFields with tumor immunotherapy have emerged as a promising therapeutic approach in clinical practice" This is a premature claim, may become true but this combination is not yet in clinical practice. Suggest rewording as "emerging" not "has emerged"
4. "There are still questions that remain unanswered, such as why TTFields only interfere with the mitosis of tumor cells" I think there is at least a partial explanation that the frequency distribution of normal vs different cancer cells is different, and could be mentioned.
5. "e immune system ’ s surveillance and continue to develop, a process known as tumor immune.." minor editing required
Comments on the Quality of English Language
Good
Author Response
Dear reviewer:
Thank you very much for your comments on this manuscript. These comments are very enlightening to us. Careful consideration has been given to all your feedback, resulting in highlighted revisions throughout the paper. Here are the responses to your suggestions:
- Response to comment: (In the plain summary section, it would be better to say "anticancer activity of TTFields" rather than "TTFields ’ anticancer activity.)
Response: Thank you for your positive comments. According to your suggestions, we have revised this sentence to read as follows: In recent years, many researchers have suggested that inducing antitumor immune responses may be one of the key mechanisms of anticancer activity of TTFields.
- Response to comment: (Abstract- Authors call it " physical therapy technology"- is better called biophysical, physical therapy means something else to a clinician.)
Response: Thank you very much for your valuable suggestion. We have carefully read the relevant literature and believe that it is more accurate to call TTFields as a biophysical therapy technology. According to your suggestions, we have revised this sentence to read as follows: Tumor treating fields (TTFields), a biophysical therapy technology that uses alternating electric fields to inhibit tumor proliferation,...
- Response to comment:("TTFields with tumor immunotherapy have emerged as a promising therapeutic approach in clinical practice" This is a premature claim, may become true but this combination is not yet in clinical practice. Suggest rewording as "emerging" not "has emerged" .)
Response: Thank you for the constructive comments. As you mentioned, TTFields combined with immunotherapy has only shown excellent results in clinical trials and has not been applied in clinical treatment. We have revised this sentence to read as follows: Treatment regimens combining TTFields with tumor immunotherapy is emerging as a promising therapeutic approach in clinical practice.
- Response to comment:("There are still questions that remain unanswered, such as why TTFields only interfere with the mitosis of tumor cells" I think there is at least a partial explanation that the frequency distribution of normal vs different cancer cells is different, and could be mentioned.)
Response: Thanks for your suggestions. We consider your viewpoint to be highly valuable, potentially contributing to our understanding of why TTFields inhibit tumor cell mitosis. According to your advice, we added a sentence as follows: The intervention frequency of alternating electric fields produces varying effects on tumor cells and normal cells. However, this viewpoint still needs more experimental data support.
- Response to comment:("e immune system’ s surveillance and continue to develop, a process known as tumor immune.." minor editing required)
Response: Thanks for your suggestions. According to your valuable suggestion, we revised a sentence as follows: However, certain neoplastic cells possess the ability to elude the vigilant scrutiny of the immune system, persisting in their development—a phenomenon denoted as tumor immune editing.

Reviewer 3 Report
Comments and Suggestions for Authors
The paper “Research progress on the mechanism of anti-tumor immune response induced by TTFields” reviews the research progress on the mechanisms of TTFields-induced antitumor immune responses and discusses their clinical significance. The objective is to offer a streamlined and precise summary to facilitate the future exploration and advancement of TTFields. The paper presents adequate references and background investigation, which is in line with the readers’ interest of cancers. However, there are still some shortcomings that need to be further improved.
Comments:
Q1. How can TTFields only induce apoptosis of tumor cells, as reflected in GA, in the complex cell composition of tumor microenvironment? At present, it seems that there is no effective methods or drugs that can kill tumor cells specifically.
Q2. Line 121, “Many experts in oncology remain skeptical of TTFields due to the lack of a precise molecular mechanism, making it difficult to maximize the therapeutic potential of TTFields in the clinic”. Is the lack of clarity of the molecular mechanism the biggest application difficulty? Or the treatment effect could not be always as expected?
Q3. If TTFields mainly show impacts on the mitotic process, could mitosis in normal cells of the body also be affected?
Q4. Most of the contents focus on the molecular changes of tumor cells and immune cells after TTFields treatment, but the underlying reasons for the correlation between TTFields and cellular changes are still unclear.
Author Response
Dear Reviewer:
Thanks very much for reviewing this manuscript. Your comments are very valuable and helpful for revising and improving our manuscript. We have studied your comments carefully and have made revisions which are highlighted in the paper. The main corrections in the paper and the responses to your comments are as follows:
- Response to comment: (How can TTFields only induce apoptosis of tumor cells, as reflected in GA, in the complex cell composition of tumor microenvironment? At present, it seems that there is no effective methods or drugs that can kill tumor cells specifically.)
Response: Thank you very much for your suggestion. This suggestion is very valuable to us. As mentioned in this manuscript, some researchers posit that alternating electric fields might trigger tumor cell apoptosis, consequently releasing tumor antigens. These antigens could be captured by dendritic cells (DC cells), activating CD8+ T cells and leading to the destruction of tumor cells. While there may not be sufficient data to fully substantiate this mechanism, these studies have shown us the possibility of this mechanism. Therefore, we chose this image (Graphical Abstract) to represent the process of TTFields inducing tumor immune response.
- Response to comment:( Line 121, “Many experts in oncology remain skeptical of TTFields due to the lack of a precise molecular mechanism, making it difficult to maximize the therapeutic potential of TTFields in the clinic”. Is the lack of clarity of the molecular mechanism the biggest application difficulty? Or the treatment effect could not be always as expected?)
Response: Thank you for the constructive comments. Your suggestion provides us with a more comprehensive reflection. As you mentioned, the lack of sufficient clinical application data is one of the reasons why TTFields is currently not widely used. According to your advice, we have revised this sentence to read as follows: Many experts in oncology remain skeptical of TTFields due to the lack of a precise molecular mechanism and sufficient data of clinical application, making it difficult to maximize the therapeutic potential of TTFields in the clinic.
- Response to comment:(If TTFields mainly show impacts on the mitotic process, could mitosis in normal cells of the body also be affected?)
Response: Thanks for your question. According to preclinical experimental data, TTFields exhibit an impact on the mitosis of tumor cells while having minimal effects on normal cells. The reasons for this phenomenon have not been fully studied yet. In response to your suggestion, we have added the following paragraph in line 118 of the article: The intervention frequency of alternating electric fields produces varying effects on tumor cells and normal cells. However, this viewpoint still needs more experimental data support.
- Response to comment: (Most of the contents focus on the molecular changes of tumor cells and immune cells after TTFields treatment, but the underlying reasons for the correlation between TTFields and cellular changes are still unclear.)
Response: Thank you for your insightful suggestion. It offers a valuable hint for our future research on the mechanism. Your perspective will be a crucial reference point in guiding our ongoing mechanism research. We have added the following paragraph in line 420 of the article: However, current research on TTFields inducing anti-tumor immune response mainly focuses on observing the changes in immune cells before and after TTFields treatment. The exploration of the correlation between TTFields and the above changes is limited. Researchers can pay more attention to the correlation between TTFields and immune response.
